# Metabolic Disturbances in Children Treated for Solid Tumors

**DOI:** 10.3390/nu11123062

**Published:** 2019-12-15

**Authors:** Ewa Barg, Joanna Połubok, Marta Hetman, Aleksandra Gonera, Olimpia Jasielska, Dorota Sęga-Pondel, Karolina Galant, Bernarda Kazanowska

**Affiliations:** 1Department of Basic Medical Sciences, Wroclaw Medical University, 50-560 Wroclaw, Poland; 2Students’ Science Society, Wroclaw Medical University, 50-560 Wroclaw, Poland; joannapolubok@gmail.com (J.P.); martha.hetman@gmail.com (M.H.); olimpia.jasielska@gmail.com (O.J.); 3Department of Pediatric Bone Marrow Transplantation, Oncology and Hematology, Wroclaw Medical University, 53-111 Wroclaw, Poland; dsega@poczta.onet.pl (D.S.-P.); karola.galant@gmail.com (K.G.); b.kazanowska@mypost.pl (B.K.)

**Keywords:** pediatric cancer, nutritional status, solid tumors, metabolic disturbances, thyroid disorders

## Abstract

Metabolic disturbances are among the most common disorders diagnosed in pediatric patients after anti-cancer therapy (ACT). The aim of our study was to evaluate the prevalence of metabolic disturbances among patients after ACT. The study group comprised 44 patients (31 boys) treated for solid tumors and 31 patients in the control group. Body weight, height, body mass index (BMI) values, lipid parameters are expressed in Standard Deviation Score (SDS), based on centile charts. Indicators of risk to atherosclerosis were calculated. Obesity/overweight was observed in one third of the patients. Hypercholesterolemia occurred in half of them, elevated tryglicerides (TG) SDS in 11, and elevated low-density lipoprotein cholesterol (LDL-C) SDS in nine of the patients. Increased levels of both cholesterol SDS and LDL SDS were found in nine patients and four of them also showed elevated levels of TG SDS. There were significant differences in lipid parameters between the sexes. Risk indicators of lipid disorders defined by statistical distances (*τ*) were determined for the study group and the control group. The sum of the risk ratios of lipid disorders in the study group was 150 times higher than in the control group. Patients after ACT require special monitoring of lipids profiles and thyroid function as they are at higher risk for dyslipidemia and atherosclerosis than healthy people.

## 1. Introduction

Improvement in the treatment of childhood malignancies has resulted in a significant increase of the 5 year survival rate among the patients. About 80% of survivors will live to adulthood and about 75% of them will experience at least one late-effect as a consequence of the disease and therapy [1,2,3]. Dyslipidemia, overweight, obesity, and metabolic syndrome are among the most common disorders diagnosed in survivors. The pathogenesis is mediated by factors related to anticancer treatment (chemotherapy, steroid therapy, and radiation), but also by additional, non-cancer agents, such as genetic, dietary, or individual [1,3,4,5]. Moreover, the higher risk of obesity may be associated with the female gender and the younger age of therapy [2,3]. Sequelae of dyslipidemia are well known and include health-related physical outcomes: high blood pressure, type 2 diabetes, metabolic syndrome, fatty liver disease, orthopedic problems and sleep apnea, but also psychological or social problems: depression, decreased self-esteem, etc. [3,4]. Simple, commonly available methods, such as height and weight measurement, BMI index, or lipid profile blood tests are very important tools in early detection of dyslipidemia, obesity, or metabolic syndrome. As such, they allow for early intervention and prevention of the consequences. Other common problems in oncological patients are various thyroid function disorders which play roles in metabolic disturbances. They may be caused by damage of the hypothalamic-pituitary axis or thyroid [6]. Hypothyroidism is diagnosed in 20% of cases after treatment of childhood cancer [7,8]. Cranial and/or spinal radiotherapy is a major contributor to both primary and secondary forms of hypothyroidism [2,7]. Overt hypothyroidism, Graves’ disease, benign nodules, and secondary thyroid malignancies may occur also after cervical radiotherapy. The role of chemotherapy is still not clear [6]. Regular control and basic hormone testing thyroid-stimulation hormone (TSH), free thyroxine (fT4) are essential in the early detection of thyroid abnormalities.

The aim of our study was to evaluate the prevalence of metabolic disturbances: lipid disorders, being overweight or obese, as well as thyroid dysfunctions among patients after anticancer treatment.

## 2. Materials and Methods 

The study group comprised 44 patients (31 boys) treated for solid tumors in the Department of Pediatric Bone Marrow Transplantation, Oncology and Hematology, Wroclaw Medical University, 3.25–16 years of age (mean 9.38 ± 3.57 and median 9.09). The patients underwent examinations evaluating the long-term effects of the anti-cancer treatment (at least one year after the treatment, 14 of them were above 5 years after the last anti-cancer therapy). The basic description of the groups is presented in Table 1. The study group was compared to the control group, which consisted of 31 healthy children at the same age, without neoplasm disease. Anthropometric measurements (body mass, height) were taken both in the study group and the control group. Body Mass Index (BMI) was calculated using the formula: weight/height^2^ (kg/m^2^). Body weight, height, and BMI values are expressed in standard deviation scores (SDS), based on centile charts, according to age and sex [9]. The standard deviation score indicates how many standard deviations an observation is above or below the mean, which is a useful way of putting data from different sources onto the same scale (Equation (1)). With the use of this statistics tool, it is possible to analyze the variability of the observed parameter over a certain period of time in a group of patients, especially those of developmental age. SDS is particularly important in the case of auxological parameters changing with age. The formula for sample-specific auxological parameters is as follows: (1)$$\mathrm{height}/\mathrm{weight}/\mathrm{BMI}\;\mathrm{SDS}\;=\;\frac{\mathrm{height}/\mathrm{weight}/\mathrm{BMI}\;\mathrm{value}-\mathrm{height}/\mathrm{weight}/\mathrm{BMI}\;\mathrm{value}\;\mathrm{for}\;50\mathrm{th}\;\mathrm{centile}}{\frac{1}{2}\times (\mathrm{height}/\mathrm{weight}/\mathrm{BMI}\;\mathrm{value}\;\mathrm{for}\;50\mathrm{th}\;\mathrm{centile}-\mathrm{height}/\mathrm{weight}/\mathrm{BMI}\;\mathrm{value}\;\mathrm{for}\;3\mathrm{rd}\;\mathrm{centile})}.$$

In the study and control groups the measurements were taken and the following laboratory parameters were tested: total cholesterol (TC) (mmol/L); low-density lipoprotein (LDL-C) (mmol/L); triglycerides (TG) (mmol/L); high-density lipoprotein (HDL-C) (mmol/L); thyroid-stimulating hormone (TSH) (uIU/ml); free triiodothyronine (FT3) (pmol/L); free thyroxine (FT4) (pmol/L); alanine aminotransferase (ALT) (U/L); aspartate aminotransferase (AST) (U/L); gamma-glutamyl transferase (GGTP) (U/L); and total bilirubin (mg/dL). Because the groups were diverse in terms of age, the results of TC, LDL-C, TG, and HDL-C levels were compared to the reference values for age and sex. They were also expressed in the values of SDS [10]. The values below normal were values ≤ −1.6 (≤10 percentile); the values above normal were values ≥ 1.6 (≥90 percentile). Based on the lipid parameters, following indicators of risk of atherosclerosis were calculated for the study group: total cholesterol/high-density lipoprotein cholesterol (TC/HDL-C), Castelli’s Risk Index (total cholesterol—HDL-C/HDL-C), LDL-C/HDL-C, and API (Atherogenic index of plasma) as the log10 (TG/HDL-C). Values above 2.5 for the ratio of LDL-C/HDL-C were considered abnormal. For the quotient of TC/HDL-C, results above 5 were considered elevated. Abnormal values for the Castelli’s RISK Index were results above 4, indicating a high predisposition to atherosclerosis. 

Risk indicators of lipid disorders defined by statistical distances (*τ*) were determined for the study and the control groups. Statistical distance is the Equation (2), wherein $E_{i}$ = median parameter of the control group; $O_{i}$ = median parameter in the study group.
(2)$$\tau =\frac{\left({\left(E_{i}-O_{i}\right)}^{2}\right)\times 100}{E_{i}}.$$

In the study group, the median in the result set of selected lipid parameters (cholesterol SDS, HDL-C SDS, LDL-C SDS, TG SDS) was compared to the corresponding results in the control group. The results of statistical distances, which may indicate an increased risk of lipid disorders development, were marked with a positive sign, whereas results showing a reduced risk of lipid disorders development received a negative sign. A statistical analysis was performed using Statistica 12. The Mann–Whitney U test was used to determine the differences between groups. The relationship between the two parameters was determined using the correlation and Pearson’s correlation coefficient. The *p*-values of less than 0.05 were considered statistically significant. The research followed the tenets of the Declaration of Helsinki. Written, informed consent was obtained from each patient before the inclusion in the study and after the aims and procedures of the research were fully explained. The study was approved by the Ethics Committee of Wroclaw Medical University (No KB-282/2014).

## 3. Results

### 3.1. Parameters Evaluated in the Study Group

Obesity or overweight was observed in 15 (34.1%) patients, nine boys and six girls. Decreased body weight was observed in two patients (4.65%). Height deficiency was found in four (9.09%) patients. Hypercholesterolemia occurred in 20 (45.46%), elevated TG SDS in 11 (25%), elevated LDL-C SDS in nine (20.45%) of the patients. Increased levels of both cholesterol SDS and LDL SDS were found in nine patients (20.45%) and four of them also showed an elevated level of TG SDS. Both Castelli’s Risk Index and the ratio of TC/HDL were increased in 11.36% of patients. A total of 29.55% patients in the study group showed an increased ratio of LDL-C/HDL-C. Overt hypothyroidism was diagnosed in five (11.36%) patients, while hyperthyroidism in one child (increased value FT4). All the results are shown in Table 2. None of the patients with CNS tumors had dysfunction of the thyroid at the moment of the examination. Only one child with hypothyroidism presented abnormal levels of lipid parameters (elevated levels of: TC SDS, TG SDS, LDL-C SDS). Other children with hypothyroidism lipid parameters were normal. In addition, elevated levels of FT3 were found in 11 (25%) children. No child had been previously diagnosed with thyroid dysfunction. Based on these studies, children with impaired thyroid function were consulted by an endocrinologist and underwent appropriate treatment for hyperthyroidism or hypothyroidism. Liver enzymes were elevated in two children and one patient had elevated total bilirubin.

### 3.2. Comparison of the Study Group with the Control Group. The Assessment of the Risk of Lipid Disorders

The children suffering from childhood cancer showed higher cholesterol SDS compared to children in the control group (mean 1.70 ± 2.51 vs. 0.52 ± 0.87, *p* = 0.027), while in case of other lipid parameters, no significant differences were observed (Table 3). The sum of the risk ratios of lipid disorders in the study group was high, suggesting an increased risk of lipid disorders in comparison to the control group. Risk ratios were positive for TC SDS, LDL-C SDS, and TG SDS. They were negative for HDL-C SDS (Table 4). The sum of the risk ratios for lipids diseases was 150.98 times higher in children with cancer than in controls. We did not observe differences in simple indicators of susceptibility to atherosclerosis (Table 4). In children in the study group, statistically lower body weight SDS was observed compared to the control group (mean 0.95 ± 1.87 vs. 1.20 ± 0.15, *p* = 0.032). Children with cancer had higher levels of FT3 and lower levels of FT4 than the controls (Table 3).

### 3.3. Differences between the Sexes

Boys accounted for 70.45% of the study group. There were no significant differences in values of body weight SDS, height SDS, and BMI SDS between girls and boys (Table 5). There were significant differences in lipid parameters between the sexes. The girls had significantly higher values of LDL SDS (mean 1.15 ± 0.76 vs. 0.70 ± 1.29, *p* = 0.033) and TG SDS (mean 1.21 ± 1.59 vs. −0.12 ± 1.56, *p* = 0.009). The indicators of susceptibility to atherosclerosis significantly differed between sexes. The girls showed higher TC/HDL-C ratio (mean 4.14 ± 0.99 vs. 3.30 ± 1.10, *p* = 0.005), Castelli’s Risk Index (mean 3.15 ± 0.99 vs. 2.30 ± 1.10, *p* = 0.005), LDL-C/HDL-C ratio (mean 2.63 ± 0.77 vs. 1.95 ± 0.91, *p* = 0.005), and the AIP (mean 0.36 ± 0.24 vs. 0.17 ± 0.26, *p* = 0.022). Table 5 presents a full description of the comparison between girls and boys.

### 3.4. Children with Obesity/Overweight

In the study group, 15 patients were overweight or obese. Seven patients (46.67%) with overweight or obesity had dyslipidemia. In the group of children with obesity or overweight, four of the 15 patients showed elevated levels of TC SDS and two of the 15 patients had elevated levels of LDL SDS. There were no significant differences in the values of lipid parameters, indicators of susceptibility to atherosclerosis, and thyroid function between patients with obesity/overweight and children with normal weight (Table 6).

### 3.5. The Correlations between the Particular Parameters

The relationship between the examined parameters was tested. In the study group, but not in the control group, a statistically significant negative correlation between body weight SDS and the HDL-C SDS (*r* = −0.35, *p* = 0.023) was found, as well as a negative correlation between BMI SDS and the HDL-C SDS (*r* = −0.36, *p* = 0.018). A positive significant correlation was found between BMI SDS and the TG SDS (*r* = 0.35, *p* = 0.021). There were no other significant correlations between anthropometric measurements, lipid parameters, or thyroid function.

## 4. Discussion

Anticancer treatment can cause many disorders in childhood cancer survivors. Many authors observed a frequent occurrence of metabolic syndrome in patients after anticancer treatment during childhood [4,11,12]. Most of the studies were performed many years after cancer treatment, mainly in adult study groups.

### 4.1. Obesity/Overweight

Disorders of body mass are common complications in the Childhood Cancer Survivor Study [7,8,13,14,15]. The most frequent problems are obesity and overweight, but being underweight [8] or having a lean mass disorder [16] also presents a significant problem. The basic method of evaluating abnormal body mass in children is comparing BMI with regard to age and gender, but some authors consider it to be inadequate in this group of patients [14]. However, in our study, we used the comparison of BMI due to its easy availability and the ability to compare the studies by other authors. In our study group, one third of the patients were obese or overweight. The most important cause of being overweight or obese in the Childhood Cancer Survivor Study is anticancer therapy, particularly radiotherapy and chemotherapy. Sklar et al. analyzed the results of 122 survivors of ALL [13]. They compared their BMI before and after the treatment. In patients who underwent radiotherapy, a significant increase in BMI at the end of therapy was noted. In the group of patients with no-radiation, no such changes were found. In addition, it was noted that with an increasing radiation dose, the proportion of patients with abnormal BMI increases [13]. Similar results were obtained by Gurney et al. and Pietila et al. in clinical survivors of brain tumors who consider radiation therapy as the most important risk factor for obesity [17,18]. Our previous studies have shown that children with soft tissue tumors at the time of diagnosis are more prone to developing overweight and obesity compared to other types of childhood cancers [19]. There are several theories explaining the causes of such complications. Sklar et al. suggested that growth hormone (GH) deficiency, occurring as a result of hypothalamus damage during radiation therapy, may play a role in causing obesity [13]. However, it cannot be the main cause as the rapid development of obesity does not correspond with the slower growth of GH deficiency [13]. Nevertheless, many authors confirmed the correlation between GH deficiency after radiotherapy and obesity or overweight [17,20]. Radiation, as treatment of brain tumors, may also directly damage hunger and satiety centers, as well as reduce their sensitivity to leptin [16]. Numerous studies have shown elevated levels of leptin in obese patients after radiotherapy [14,16]. However, the main cause is still unclear. We do not know if it is a compensatory increase secretion or increased amount of fat cells in patients with obesity [16]. Another mechanism was proposed by Willson et al. [14]. They discovered a correlation between genes (polymorphisms: rs4971486, rs4530610, rs2923762, rs35669975, rs12073359) and obesity in patients after radiotherapy [14]. On the other hand, Van Dongen-Melman et al. showed no association of radiotherapy with obesity [21]. Steroids are also a significant cause of obesity in patients after anticancer treatment. Van Dongen-Melman et al. compared the group of patients who received different variants of anticancer treatment [21]. At the end of treatment, the highest percentage of obese patients was in the group treated with dexamethasone without radiotherapy, while four years later, in patients treated with dexamethasone and prednisolone without the radiotherapy. Authors showed that steroids, through their adverse effects on hormones, have a greater impact on body mass disorders than radiotherapy and can cause obesity during therapy and support its growth after cessation of treatment [21]. This is confirmed by other studies [14]. In our study group, all of the overweight patients were treated with dexamethasone. Diet and physical activity also have a significant impact on the development of obesity [22]. On the other hand, in a study of craniopharyngioma survivors, Harz et al. found no increased food intake in comparison to the control group. However, they noted decreased physical activity in patients and demonstrated that it has a dominant influence on body weight in this group [23]. In our study, we found no significant difference in the prevalence of obesity between boys and girls. Moreover, age at diagnosis had no effect on BMI. Many other authors arrived at the same results [17,21]. Nevertheless, Gurney et al. revealed a higher incidence of obesity in girls and in younger patients [18]. In our study, the problem of increased BMI affected one third of the patients after anticancer treatment. Similar results were obtained by Wilson et al. [14]. Obesity (BMI > 30) was found in 36.5% of male and 36.5% female patients [14]. Likewise, in their research, Gurney et al. noted BMI > 30 in 15% of survivors [18]. Obesity in children has several serious consequences, such as increased incidence of chronic diseases and mortality [24]. It is a risk factor for cardiovascular diseases, it also causes a reduction in physical activity [25], and can lead to metabolic syndrome [20], especially lipids disorders and insulin resistance [17]. This is confirmed in our findings, in which lipids disorders are not only direct complication of anticancer therapy, but also the result of abnormal body weight. Obesity is an important problem in childhood cancer survivors because of the prevalence and possible complications. It is necessary to monitor body mass, as well as further research the effects of anticancer therapy on the development of obesity and possible prevention methods.

### 4.2. Lipids Profile

In our study, we often observed dyslipidemia. Hypercholesterolemia occurred in about 50% of the patients, while elevated triglycerides SDS and elevated LDL-C SDS, in one fourth of the patients. The common abnormalities in lipids profile in our study group are concerning, especially because our study group consisted of young children (aged 3.25–16 years old). In our analysis, we also included other factors that could have a direct impact on lipids profile, for example, disturbance in thyroid function or liver damage. Neither were significant problems in our study group as only 4.55% of the patients had elevated liver enzymes, 11.36% had hypothyroidism, and only 2.27% had hyperthyroidism. The above-mentioned results suggest that the occurrence of dyslipidemia is not solely caused by thyroid disfunction or liver damage. Other pathomechanisms should also be taken into consideration. Furthermore, we did not observe statistically significant differences in lipids parameters between obese/overweight children (30.24% of the study group) and children with proper body weight. Seven (46.67%) children with obesity/overweight had dyslipidemia. In the study group with proper body weight, seventeen (58.62%) had elevated at least one parameter of lipids fraction. Our results strongly suggest that dyslipidemia is a common abnormality also in children with proper body weight. Radiotherapy, especially cranial radiotherapy (CRT), may induce metabolic syndrome, which is commonly associated with dyslipidemia. Patients who received CRT more frequently had components of metabolic syndrome [2,17,20]. The results of our study suggest that dyslipidemia occurs frequently also in children with cancer. We observed that childhood cancer survivors had higher index of risk of lipids diseases (*τ*) than the control group. The sum of the risk ratios of lipids diseases was 150.98 greater in children with cancer than in the healthy control group. Patients had a higher risk for elevated TC SDS (*τ* = 180.3), TG SDS (*τ* = 10.03), and LDL-C SDS (*τ* = 8.82) than the control group. Our results show that the highest risk for lipid diseases is caused by a higher cholesterol level. We did not observe a higher risk for low HDL-C SDS in our patients than in the control group. Other authors had similar results, i.e., dyslipidemia occurred more often in childhood cancer survivors than in the control groups [4,11,12,26,27]. In most of the studies, the control groups were healthy siblings. Therefore, it seems that the metabolic status of patients should be monitored after the completion of anti-cancer therapy [28].

### 4.3. Atherosclerosis

Atherosclerosis begins in childhood, especially if children have the following risk factors: dyslipidemia, hypertension, obesity, or insulin resistance [29]. Mediastinal radiation can cause coronary artery disease (CAD). After irradiation, coronary endothelial cells are probably damaged [30,31]. This damage causes fibrointimal hyperplasia which leads to thrombus formation and lipid deposition [2]. A high dose of mediastinal radiation (>20 Gray (Gy)) could be considered a risk factor for CAD. The patients who received higher dose of radiotherapy had six to seven times higher risk for CAD than the patients who received a lower dose of radiation [32]. In our study, in order to evaluate the risk of atherosclerosis, we used indicators for susceptibility to atherosclerosis, for example, Castelli’s Risk Index or LDL-C/HDL-C ratio. LDL-C/HDL-C ratio was elevated in one-third of the patients and Castelli’s Risk Index—in 11.36%. We did not observe differences in values of indicators for the susceptibility of atherosclerosis between our study group and controls. Our results suggest that the risk of atherosclerosis is high in children with cancer because of the high prevalence of an increased ratio of LDL-C/HDL-C. Low HDL-C and high LDL-C are common risk factors for cardiovascular diseases.

### 4.4. Thyroid Function

Among other complications of anticancer therapy, particularly important are thyroid disorders because of their prevalence and consequences that can result in a developing organism [33]. Radiation therapy is the most important risk factor for damage to the thyroid [7]. The biggest damages are caused by neck, craniospinal, chest [2], and total body radiation [34]. Moreover, the survivors of Hodgkin disease, brain tumors, and leukemia are particularly vulnerable to complications. Duration of treatment and the dose of radiation play an important role—the higher the dose and the longer the duration, the more likely are side effects [7,33]. The role of chemotherapy is still not clear in damage to the thyroid. While many studies recognize this association with damage to the thyroid [35], other authors did not record this phenomenon [6]. Caglar et al. analyzed the results of 120 patients who are survivors of different types of childhood cancers. They divided patients into two groups—the first treated with chemotherapy alone and the second with a combination of chemotherapy and radiotherapy. Hypothyroidism was found only in the group treated with radiotherapy. Other complications, such as thyroid nodules and tumors, were also more common in this group. Chemotherapy had no effect on the function and complications of thyroid [6]. Among disorders of the thyroid, the most common is hypothyroidism [2,34]. This may be the result of direct thyroids damage by neck radiation (primary hypothyroidism) [33,34] or indirect, resulting in deficiency of TSH, after craniospinal radiation (central hypothyroidism) [6]. It is debatable which type prevails. Surprisingly, some authors noted a prevalence of primary hypothyroidism in children with brain tumors treated with craniospinal radiation [6,33,35]. This could be the result of damage to the thyroid by some degree of radiation scatter [6]. In contrast, Ramanauskienė et al.’s studies revealed that 63.6% of the children were suffering from central hypothyroidism, while 36.4% had primary hypothyroidism [7]. In fact, TSH deficiency is less common than other hormone deficiencies and often occurs subclinically [34]. Some authors noted isolated cases of coexistence of central and primary hypothyroidism [8]. Hyperthyroidism is a rare complication of radiotherapy, although it occurs more frequently than in the comparable control groups [2,33]. Other complications are benign nodules, Graves’ disease, and cancer—caused by radiotherapy and dependent on the dose of radiation [2,6]. Thyroid function should be monitored in patients after anticancer treatment to avoid overlooking the long-term consequences of its damage. Younger age at diagnosis and female sex are considered independent risk factors for hypothyroidism [35,36,37].

## 5. Conclusions

The results of our study explicitly emphasize the problem of lipids abnormalities in childhood cancer survivors after anticancer treatment. This group requires special monitoring of lipid profiles as they are at higher risk for dyslipidemia and atherosclerosis than healthy people. The thyroid function should also be monitored, especially in the case of radiation therapy. The results obtained from the study encourage its further extension to a larger group of cancer survivors.

## Figures and Tables

**Table 1 nutrients-11-03062-t001:** The basic characteristics of the study group.

Cancer	*n*	Boys/Girls	Age at Diagnosis (Years)	Age at Examination (Years)
Soft tissue tumor	27	19/8	6.90 ± 4.54	9.73 ± 3.64
Bone tumor	4	1/3	12.25 ± 3.40	14.49 ± 3.53
Neuroblastoma	5	5/10	5.81 ± 5.22	8.11 ± 3.20
Central nervous system tumor	3	3/0	4.67 ± 3.31	8.12 ± 3.57
Germ cell tumor	5	3/2	7.39 ± 8.51	10.71 ± 2.63

**Table 2 nutrients-11-03062-t002:** Parameters evaluated in the study group.

Parameter	*n*	Mean Value	Lowered (%)	Normal (%)	Elevated (%)
Body weight SDS	43	0.95 ± 1.87	1 (2.32%)	28 (65.11%)	14 (32.56%)
Height SDS	44	0.03 ± 1.26	4 (9.09%)	40 (90.91%)	0 (0%)
BMI SDS	43	1.33 ± 2.06	2 (4.65%)	28 (65.11%)	13 (30.24%)
TC SDS	44	1.67 ± 2.51	1 (2.27%)	23 (52.27%)	20 (45.46%)
LDL-C SDS	44	0.8 ± 1.17	−	35 (79.55%)	9 (20.45%)
TG SDS	44	0.27 ± 1.67	2 (4.55%)	31 (70.45%)	11 (25%)
HDL–C SDS	44	0.18 ± 1,41	3 (6.82%)	36 (81.82%)	5 (11.36%)
TC/HDL-C	44	3.55 ± 1.13	−	39 (88.64%)	5(11.36%)
Castelli’s Risk Index	44	2.55 ± 1.13	−	39 (88.64%)	5 (11.36%)
LDL-C/HDL-C	44	2.15 ± 0.92	−	31 (70.45%)	13 (29.55%)
TSH (uIU/mL)	44	2.79 ± 2.09	0 (0%)	39 (88.64%)	5 (11.36%)
FT4 (pmol/L)	44	15.38 ± 2.49	0 (0%)	43 (97.73%)	1 (2.27%)
FT3 (pmol/L)	44	7.37 ± 2.2	0 (0%)	33 (75%)	11 (25%)
ALT (U/L)	44	15.7 ± 8.49	−	43 (97.73%)	1 (2.27%)
AST (U/L)	44	25.02 ± 7.15	−	42 (95.45%)	2 (4.55%)
GGTP	44	16.23 ± 5.09	−	44 (100%)	0 (0%)
Total bilirubin	44	0.49 ± 0.25	−	43 (97.73%)	1 (2.27%)

SDS—standard deviation score; BMI—body mass index; TC—total cholesterol; LDL-C—low-density lipoprotein cholesterol; TSH—thyroid-stimulation hormone, FT4—free thyroxine, FT3—free triiodothyronine, TG—triglycerides; HDL-C—high-density lipoprotein cholesterol; ALT—alanine aminotransferase; AST—aspartate aminotransferase, GGTP—gamma-glutamyl transferase.

**Table 3 nutrients-11-03062-t003:** Parameters in the study and control groups.

Parameter	Study Group *n* = 44	Control Group *n* = 31	*p* Value
Body weight SDS	0.95 ± 1.87	1.20 ± 0.15	**0.032**
Height SDS	0.03 ± 1.26	−0.31 ± 2.22	0.369
BMI SDS	1.33 ± 2.06	0.33 ± 2.13	**0.067**
TC SDS	1.70 ± 2.51	0.52 ± 0.87	**0.027**
LDL-C SDS	0.83 ± 1.17	0.60 ± 1.05	0.554
TG SDS	0.27 ± 1.67	0.52 ± 3.43	0.429
HDL–C SDS	0.18 ± 1.41	−0.08 ± 1.89	0.272
TC/HDL-C	3.55 ± 1.13	3.70 ± 1.42	0.663
Castelli’s Risk Index	2.55 ± 1.13	2.70 ± 1.42	0.663
LDL-C/HDL-C	2.15 ± 0.92	2.25 ± 1.11	0.739
AIP	0.23 ± 0.27	0.19 ± 0.32	0.456
TSH (uIU/mL)	2.79 ± 2.09	2.31 ± 1.13	0.352
FT4 (pmol/L)	15.38 ± 2.49	16.97 ± 3.11	**0.005**
FT3 (pmol/L)	7.37 ± 2.2	5.92 ± 1.17	**0.001**

SDS—standard deviation score; BMI—body mass index; TC—total cholesterol; LDL-C—low-density lipoprotein cholesterol; TG—triglycerides; HDL-C—high-density lipoprotein cholesterol; AIP—Atherogenic Index of Plasma; TSH—thyroid-stimulation hormone, FT4—free thyroxine, FT3—free triiodothyronine; *p* Value bold—statistically significant values.

**Table 4 nutrients-11-03062-t004:** Risk ratios of lipid disorders in the study and the control groups.

Parameter	Median		The Risk Ratio of Lipid Disorders (Statistical Distance)
	Study Group	Control Group	
TC SDS	1.43	0.49	180.3
LDL-C SDS	0.83	0.60	8.82
TG-SDS	−0.17	−0.36	10.03
HDL-C SDS	−0.03	−0.54	−48.17
**Sum (*τ*)**			**150.98**

SDS—standard deviation score; TC—total cholesterol; LDL-C—low-density lipoprotein cholesterol; TG—triglycerides; HDL-C—high-density lipoprotein cholesterol.

**Table 5 nutrients-11-03062-t005:** Comparison between girls and boys in the study group.

Parameter	*n*	Girls *n* = 13	Boys *n* = 31	*p* Value
Body weight SDS	43	1.46 ± 1.83	0.73 ± 1.87	0.195
Height SDS	44	0.20 ± 1.03	−0.04 ± 1.35	0.857
BMI SDS	43	1.74 ± 2.25	1.16 ± 1.99	0.284
TC SDS	44	2.29 ± 1.8	1.45 ± 2.74	0.068
LDL-C SDS	44	1.15 ± 0.76	0.70 ± 1.29	**0.033**
TG SDS	44	1.21 ± 1.59	−0.12 ± 1.56	**0.009**
HDL–C SDS	44	−0.48 ± 1.29	0.45 ± 1.39	0.068
TC/HDL-C	44	4.14 ± 0.99	3.30 ± 1.10	**0.005**
Castelli’s Risk Index	44	3.15 ± 0.99	2.30 ± 1.10	**0.005**
LDL-C/HDL-C	44	2.63 ± 0.77	1.95 ± 0.91	**0.005**
AIP	44	0.36 ± 0.24	0.17 ± 0.26	**0.022**
TSH (uIU/ml)	44	2.01 ± 1.0	3.13 ± 2.33	**0.066**
FT4 (pmol/l)	44	15.51 ± 3.35	15.33 ± 2.09	0.139
FT3 (pmol/l)	44	8.17 ± 2.89	7.04 ± 1.79	0.767

SDS—standard deviation score; BMI—body mass index; TC—total cholesterol; LDL-C—low-density lipoprotein cholesterol; TG—triglycerides; HDL-C—high-density lipoprotein cholesterol; AIP—Atherogenic Index of Plasma; TSH—thyroid-stimulation hormone, FT4—free thyroxine, FT3—free triiodothyronine; *p* Value bold—statistically significant values.

**Table 6 nutrients-11-03062-t006:** Comparison between overweight/obesity and normal body weight group among study group.

Parameter	*n*	Overweight/Obesity *n* = 15	Normal Body Weight *n* = 28	*p* Value
TC SDS	44	0.92 ± 1.77	2.18 ± 2.78	0.150
LDL-C SDS	44	0.53 ± 0.81	1.03 ± 1.3	0.221
TG SDS	44	0.50 ± 1.61	0.20 ± 1.72	0.532
HDL–C SDS	44	−0.17 ± 1.61	0.35 ± 1.32	0.194
TC/HDL-C	44	3.63 ± 1.08	3.55 ± 1.18	0.483
Castelli’s Risk Index	44	2.63 ± 1.08	2.55 ± 1.18	0.584
LDL-C/HDL-C	44	2.18 ± 0.82	2.16 ± 0.98	0.584
AIP	44	0.30 ± 0.28	0.20 ± 0.26	0.314
TSH (uIU/ml)	44	2.58 ± 1.21	2.91 ± 2.48	0.750
FT4 (pmol/l)	44	15.71 ± 3.34	15.11 ± 1.93	0.909
FT3 (pmol/l)	44	7.77 ± 2.85	7.13 ± 1.83	0.468

SDS—standard deviation score; TC—total cholesterol; LDL-C—low-density lipoprotein cholesterol; TG—triglycerides; HDL-C—high-density lipoprotein cholesterol; AIP—Atherogenic Index of Plasma; TSH—thyroid-stimulation hormone, FT4—free thyroxine, FT3—free triiodothyronine.

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
