# Peer review of "Metabolic Disturbances in Children Treated for Solid Tumors"

_nutrients, 2019, doi:10.3390/nu11123062_

Round 1

Reviewer 1 Report

The authors have addressed the issues that I raised in my initial review, I thank them for doing so. 

Author Response

Thank you for your review.

Reviewer 2 Report

It is an important study addressing health issues of vulnerable population. Overall, it is a good paper, I just have few comments:

In Results section, page 4, line 141, the use of children was confusing, readers may think the authors are referring to the control group. Authors may want to replace "children" with "patients". All Tables should be self explanatory, need to spell out all abbreviations at the end of each Table.  Table 3, need to show differences between study group and control group in thyroid hormones.  Table 6 title, need to add "among study group" page 7 line 236, need please to replace "in the group of children" with "study group" to prevent confusion

Author Response

Thank you for review.
1. we replace "children with patients".
2. All tablets have self-explanatory and spell out all abbreviations at the end of each Table.
3. In table 3 we show differences between study group and control group in thyroid hormones
4. In Table 6 title, we added "among study group"
5. In page 7 line 236, we replaced "in the group of children" with "study group"

This manuscript is a resubmission of an earlier submission. The following is a list of the peer review reports and author responses from that submission.

Round 1

Reviewer 1 Report

The Authors of the paper : “Metabolic disturbances in children treated for solid tumors “ concludes that patients ACT require special monitoring of lipids profiles and thyroid function as they are at higher risk for dyslipidemia and atherosclerosis than healthy people. According to the Authors the pathogenesis of such metabolic disturbances would be mediated by factors related to anticancer treatment (chemotherapy, steroid therapy and radiation), but also by additional, non-cancer agents, such as genetic, dietary or individual factors. Other common problems in oncological patients would be thyroid function disorders which play roles in metabolic disturbances. Their cause may be the damage of the hypothalamic-pituitary axis and/or thyroid gland . Hypothyroidism is diagnosed in 20% of cases after treatment of childhood cancer. Cranial and/or spinal radiotherapy is a major contributor to both primary and secondary forms of hypothyroidism. I observe that : 1) The patients underwent examinations evaluating the long-term effects of the anti-cancer treatment (at least one year after the treatment). However, it is not clear the exact time after the treatment for each tumor case. 2) The Authors considered patients with heterogeneous types of tumors, in their series the most of them had soft tissue tumors, this should be considered the most significant group of patients. They included CNS tumors, that have completely different clinical aspects which could specifically involve metabolic effects ( with damage of the hypothalamic-pituitary axis). I consider the subject of the paper very interesting, but the series of patients look heterogeneous ( considering the type of tumors). Moreover, cases with CNS involvement should be considered in a different distinct group of cases. I think that the paper is suitable for publications with major changes, considering the different tumors types and the different protocols of treatment.

Reviewer 2 Report

The manuscript "Metabolic disturbances in children treated for solid tumors"  looks at the metabolic disturbances of children undergoing treatment for childhood cancers.  The study is validated by comparing the findings to that of a group of children not undergoing treatment.  The problem with any of these studies is that there are not large numbers in the study group.  However there were 44 in the study group so it does make the study valid.  It would have been better had there been more that 13 girls in this group, which would have given the study greater statistical significance.  The authors found that lipid disturbances were much greater in the cohort being teated for their cancers than that of the healthy group.  This finding should prompt other studies involving larger cohorts to confirm this observation and to then be used to see if treatments can be instigated to reduce these lipid profiles.  It is in this basis I have rated this manuscript as being significant.

Having read the manuscript I have the following comments:

 Can the authors list the city and country where they work (lines 6-10), as well as their email addresses as per MDPI journal requirements. In the discussion when stating an author's name it is best to cite the reference number at the end of that sentence rather than some sentences later. Line 186 you mention numerous studies about Leptin but no references are cited, please add these. Line 189 it is not clear what you are referring to about SNP's, to which gene(s) are you referring to, it is not clear Line 198 you refer to multiple studies but only cite one reference either add another reference or change your statement, as studies implies 2+ references. Lines 219 and 227, either use British or American spelling but not both as dyslipidaemia is spelt both ways. In citing numbers with decimal places use a full stop and not a comma, eg. Line 239 and elsewhere it should read 150.98 not 150,98 In the reference list either cite the full name of journals or their abbreviated titles but not both.  Also list journal titles in italics.